# Stakeholders' views of supporting asthma management in schools with a school-based asthma programme for primary school children: a qualitative study in Malaysia

Siti Nurkamilla Ramdzan ![ORCID] ,[1,2] Ee Ming Khoo,[1] Su May Liew,[1] Steve Cunningham,[2] Hilary Pinnock,[2] on behalf of RESPIRE collaborators

in December 2021

¹Department of Primary Care Medicine, Universiti Malaya, Kuala Lumpur, Malaysia
²NIHR Global Health Research Unit on Respiratory Health (RESPIRE), The University of Edinburgh Usher Institute of Population Health Sciences and Informatics, Edinburgh, UK

**Correspondence to**
Professor Hilary Pinnock;
Hilary.Pinnock@ed.ac.uk

## ABSTRACT

**Objective** The WHO Global School Health Initiative aimed to improve child and community health through health promotion programmes in schools, though most focus on preventing communicable disease. Despite WHO recommendations, no asthma programme is included in the Malaysian national school health service guideline. Therefore, we aimed to explore the views of school staff, healthcare professionals and policy-makers about the challenges of managing asthma in schools and the potential of a school asthma programme for primary school children.

**Design** A focus group and individual interview qualitative study using purposive sampling of participants to obtain diverse views. Data collection was guided by piloted semistructured topic guides. The focus groups and interviews were audiorecorded, transcribed verbatim and analysed using inductive thematic analysis. We completed data collection once data saturation was reached.

**Setting** Stakeholders in education and health sectors in Malaysia.

**Participants** Fifty-two participants (40 school staff, 9 healthcare professionals and 3 policy-makers) contributed to nine focus groups and eleven individual interviews.

**Results** School staff had limited awareness of asthma and what to do in emergencies. There was no guidance on asthma management in government schools, and teachers were unclear about their role in school children's health. These uncertainties led to delays in the treatment of asthma symptoms/attacks, and suggestions that an asthma education programme and a school plan would improve asthma care. Perceived challenges in conducting school health programmes included a busy school schedule and poor parental participation. A tailored asthma programme in partnerships with schools could facilitate the programme's adoption and implementation.

**Conclusions** Identifying and addressing issues and challenges specific to the school and wider community could facilitate the delivery of a school asthma programme in line with the WHO School Health Initiative. Clarity over national policy on the roles and responsibilities of school staff could support implementation and guide appropriate and prompt response to asthma emergencies in schools.

### Strengths and limitations of this study

► One of a few studies on school asthma programmes conducted in low-income and middle-income countries.
► Various backgrounds contributed to the diverse views of participants.
► Some loss of meaning may have occurred during the translation of Malay quotes to English, but the analysis was conducted in the source language.
► Participants with less experience on the topic may have contributed less to the focus groups though they were provided with opportunities to provide their views.

## INTRODUCTION

The Global School Health Initiative launched by the WHO in 1995 aimed to improve child, adolescent and community health through health promotion programmes in schools.[1] Globally, the compulsory school years offer an 'easy entry point' for embedding healthy behaviours at a young age and, by extension, influencing families.[2] Health promotion programmes (eg, deworming, vaccination, water, sanitation, hygiene) have been successfully implemented in low-income and middle-income countries (LMICs) through this initiative.[2] Non-communicable diseases (NCDs) are an increasing challenge in all economies, though successful NCD programmes have typically been focused in high-income countries.[2 3]

Asthma is the the most common NCD in children with a global prevalence of 11.6%.[4] Children with poor asthma control have symptoms of wheeze, cough and difficulty breathing that affect their daily activities.[5] Good self-management of asthma improves symptoms and reduces the risk of attacks,[6] but

for primary school children, self-management of asthma requires support from their social network including their parents, school and teachers.[7] Young children can spend up to half their waking hours in schools, and school staff could help students develop asthma self-management skills.[8–10] Furthermore, attacks can occur unexpectedly, and schools need plans for responding appropriately and promptly to asthma symptoms.[5 11]

The WHO endorses the role of school in supporting children with asthma and recommends that asthma should be an essential school health service in all countries.[3 12 13] School-based asthma self-management educational interventions that were theory-driven, involved parent, included children's satisfaction and done outside children's free time have been shown to improve asthma control, reduce school absenteeism and asthma attacks,[14–16] though few were conducted in LMICs.[14 15] These programmes are complex interventions that can be successful if tailored to the target population,[17 18] and adapted to the context.[19] The WHO has also highlighted the importance of policy and organisation-level collaboration and engagement with parents, students and teachers.[2]

In Malaysia, an estimated 9% of children have asthma,[20 21] at least half of whom are poorly controlled.[20 22] Despite this morbidity, the national school health service guideline does not include an asthma programme,[23] and prior qualitative study conducted among primary school children with asthma and their parents revealed a perception of poor support for asthma care in schools.[7 24]

We, therefore, aimed to explore the views of school staff, healthcare professionals and policy-makers about the challenges of managing asthma in schools and the potential of a school asthma programme for primary school children to improve care.

## METHODS

This mixed focus group and individual interview qualitative study was conducted between May and December 2019. We obtained permission from the schools, the Malaysian Ministry of Health and the Ministry of Education to conduct the study. All participants provided their written informed consent.

### Context and sampling

About 98% of Malaysia's primary education takes place in government schools under the Ministry of Education.[25] In these schools, there are no school nurses and the healthcare of school children is provided by the Ministry of Health in a form of a school health team composed of family medicine specialists, medical officers and nurses, who provides general health screening and education at each district.[23] We conducted the study in the central region of Peninsular Malaysia and purposively sampled school staff (teachers and administrative staff) from five government schools (two Malay, two Chinese and one Indian), to represent the three main ethnic groups and

**Table 1** Demographic characteristics of participants (N=52)

| Demographic | n (years (mean)) |
|---|---|
| Age range: years (mean) | 23–58 (42) |
| Female | 46 |
| Male | 6 |
| Working experience: years (mean) | 1–35 (17) |
| Education status | |
| Primary/secondary | 0 |
| Tertiary | 52 |
| Employment background | |
| Healthcare professional | 9 |
| School staff | 40 |
| Policy-maker | 3 |
| Ethnicity | |
| Malay | 20 |
| Chinese | 18 |
| Indian | 14 |
| Background of participants | |
| Personal history of asthma | 8 |
| Family history of asthma | 11 |
| Certified healthcare professional | 13 |

diverse cultures in Malaysia.[25] We also purposively invited participants from a non-governmental organisation and a private school, healthcare professionals including the school health team and policy-makers to obtain views from the wider education community. Potential participants were invited individually or via their organisation by providing them with a participant information sheet.

### Data collection

We arranged focus groups to enable peer interaction supplemented with individual interviews to accommodate the availability and preference of participants.[26 27] We collected basic sociodemographic data, for example, age, gender, profession and asthma experience (online supplemental 1, with variables reported in table 1), and developed piloted semi-structured topic guides (online supplemental 2) based on the biopsychosocial model and previous literature to stimulate discussion.[7 24 28 29] Different topic guides containing specific questions relevant to different roles were used for different professions. The core topics addressed were asthma experience and awareness, current school asthma management practices and ideas for improvement. All the interviews and focus groups were conducted either in Malay (the national language of Malaysia) or English and moderated by SNR, a researcher who is proficient in both languages. We anticipated up to 50 participants and concluded data collection when there were no new themes emerging based on our core topics. The focus groups and interviews were audiorecorded, transcribed verbatim and analysed using Nvivo software V.11.

## Language and translation

The transcripts were analysed in the source language to reduce loss of meaning with translation. Aided by the field notes, the Malay transcripts were translated to English by experienced bilingual translators, and SNR checked the translated transcripts for semantic and contextual equivalence. The translated transcripts were used for data sharing, for example, the use of quotes to illustrate the findings in this report.

## Data analysis

We used inductive thematic analysis, a systematic but flexible approach that is often used to inform intervention development and evaluation.[26 30] SNR read the transcripts multiple times for data familiarisation and developed initial codes according to the research question. Constant comparison and merging of initial codes allowed the development of subcodes, subthemes and themes, and we then constructed a schema to describe the relationship between themes. Memos, field notes and discussions with the research team aided data interpretation.

### Reflexivity and interpretation

SNR who undertook data collection and coding, is an academic family physician with more than 15 years of working experience in Malaysia. At the beginning of the focus groups/interviews, SNR explained that the main purpose of the study was to improve the care of children with asthma. She emphasised her position as a researcher in understanding participants' views without judgements. The analysis and results were discussed with the multi-cultural team consisting of primary care physicians and a paediatrician from the UK and Malaysia, to ensure balanced interpretation. We used a pragmatic and constructivist approach but constantly reminded ourselves to be open to any possible emerging theory.

### Patient and public involvement

We piloted the topic guides among teachers and health professionals when designing the study. They provided feedback on the approach and questions in the topic guides. We shared the preliminary findings of our study with some participants and other stakeholders in stakeholder engagement meetings.

## RESULTS

We recruited 52 participants consisting of 40 school staff (including three from a private school), nine healthcare professionals (five from the school health team), three policy-makers including a representative from the teachers' professional association. We conducted nine focus groups (up to six participants for each group; duration up to 40 min) and 11 individual interviews (duration up to 45 min). Table 1 shows the characteristics of the participants and table 2 details the participants in each focus group. The participants' age ranged from 23 to 58 years and their working experience ranged 1–35 years. A

| Table 2 | Details of participants in each focus group |
|---|---|
| **Focus group** | **Participants** |
| 1 | 5 teachers |
| 2 | 5 teachers |
| 3 | 4 nurses and 1 nurse assistant |
| 4 | 3 teachers and 1 clerk |
| 5 | 4 teachers |
| 6 | 6 teachers |
| 7 | 3 teachers and 1 clerk |
| 8 | 6 teachers |
| 9 | 2 school nurses and 1 school administrator |

quarter of them (14) had a personal history of asthma or a family member with asthma.

## Summary of findings

Figure 1 illustrates the inter-relationship of the themes and subthemes. Four themes were constructed to inform the development of a school asthma programme; the first highlighted the problems perceived by participants, the second reported the suggested solutions and the two following themes addressed the challenges and enablers.

Quotes describing the themes are provided in text boxes and additional quotes are provided in boxes 1 and 2.

## Problems
### Variable awareness of asthma

The awareness of asthma among the participants depended on their personal experience of having asthma or helping others to self-manage asthma, for example, family members or students, and/or if they had received asthma training or education (box 1). Participants with experience or training could describe symptoms and were familiar with the treatment for asthma symptoms/attacks. Other participants were less able to contribute to the discussion about asthma symptoms, treatment and possible complications. They had several misconceptions, for example, 'children with asthma cannot do physical activities', and 'attacks do not occur suddenly in a well child'.

> Moderator: What happens if the kid has asthma, but he/she says that he/she can do it (physical activities)?
>
> C2: We don't let them do it. (FGD2_SS, school staff)

> E2: I think if the child is having asthma (symptoms), it's best for the parents to bring their child to see the doctor. Can an asthma attack suddenly occur when the child is well?
>
> Moderator: Yes.
>
> E2: I don't know about that. (FGD2_SS, school staff)

| **Problems** |
| --- |
| ♦ Variable awareness of asthma<br>♦ Reliance on generic schools' medical plans<br>♦ No communicated individual asthma action plans |

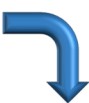

| **Suggested solutions** |
| --- |
| ♦ Asthma education and asthma plan at school<br>♦ Improve asthma self-management of children with support of parents and teachers<br>♦ Improve communication between parents, school & healthcare providers |

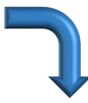

| **Implementation challenges** |
| --- |
| ♦ Heavy workload of teachers and school health team<br>♦ Poor parental participation<br>♦ Diverse individual capabilities and needs |

| **Implementation enablers** |
| --- |
| ♦ Simple, engaging and tailored programme approaches<br>♦ Good partnership with schools<br>♦ Multi-organisation involvement |

**Figure 1** Themes and subthemes of participants' views of asthma and school-based asthma intervention.

### Reliance on generic medical school plans

All participants stated that there was no specific asthma school plan in government schools (box 1), so that staff followed the general advice (eg, in circular letters from authorities) and their generic school plan for any illness or medical emergency. If a child was unwell, teachers would contact the parents to fetch them from school, and the parents decided to either treat the child themselves or to seek medical attention. In an emergency, teachers would take the child for medical care if it was clear a child needed urgent treatment, or their parent was taking too long to get to school.

> Usually what we do, first, is to call their parents. If their parents said, 'okay, can bring them to the clinic'. Then we will bring them to the nearest clinic. We will get a permission letter from the office. (A5_FGD 5_SS, teacher)

---

**Box 1    Additional quotes in themes illustrating problems**

**Variable asthma awareness**
'As for me, I would observe them, cool them down since I don't have any experience. Basically, I lacked the knowledge on how to treat asthma. So, that's the only way I can help.' (RO4_FGD4_SS, school staff)
'At night if I cough, I will take the blue inhaler and usually, if I use the inhaler and it doesn't work… I will go to the emergency department to take nebuliser.' (A14_FGD 4_SS, school staff with asthma)
**Reliance on generic medical school plans**
'If the parents contacted me and said that my child can use this and know how to use this, I will tell the parents that it is the parents' and the child's responsibility to use the medication… You brought it to school and use it on your own. If a child has a more serious condition, we will allow the parents to enter the school to give medication according to the timing.' (E2_FGD 2_SS, teacher)
'A5: Sometimes, when we give treatment to a student.
C5: parents blamed the teacher.
HA5: we are only afraid that something worse can happen. For example, we give medication, suddenly the child's condition gets worse.' (FGD 5_SS, teachers)
**No communicated individual asthma action plan**
'I have a student with asthma, her mother said she has an inhaler, but she does not bring it to school. I've never seen it.' (D_FGD 1, teacher)

---

**Box 2    Additional quotes illustrating themes for solutions, challenges and enablers**

**Suggested solutions**
'For improvement, we should ask parents to provide the asthma book together with their immunisation records. So, we can monitor their asthma appointment.' (FA3_FGD 3, school health team)
**Implementation challenges**
'We have many programmes, like the health-promoting programme, vaccination programme, the teacher also requested us to finish as soon as possible because they had to go through their education programme.' (K2_IDI 2, school health team)
**Implementation enablers**
'It has to be concise and simple. I think half an hour to an hour is enough. More than that, they (children) won't be able to concentrate.' (AN6, F.G.D. 6, teacher)
'SA6/RA 6: Malay can understand.
AN6: Even if they cannot understand, a teacher will help to explain in Tamil. No problem.' (F.G.D. 6, teachers)
'We have a parent and community organisation, and the school parent and teacher organisation who can help to promote the programme. Parents can know about the programme from them, and they can come.' (T6_IDI 6, teacher)

The teachers were unclear about their role in schoolchildren's health. The circular letters stated that staff members were not allowed to give medication to students, but this was interpreted differently according to their personal views and the school management plan. Some teachers interpreted the circular letters to only apply to non-prescribed medication and would act in loco parentis and give treatment to children only if they had clear instructions from their parents. A few teachers felt able to extend this to supervising prescribed medication such as an inhaler (although not being sure of the technique). Other teachers would not want to handle medication at all as they viewed medication use as the responsibility of parents and children. They feared their action might worsen the student's condition, for example, giving the wrong dose, causing side effects of medication, being blamed by parents as they were not trained to manage medical conditions.

> Sometimes parents called the teachers and said, 'my child has an inhaler in his bag. Please can you supervise him'… Usually children will know how to use it. We just supervise when he is using it. (SA6_FGD 6_SS, teacher)

### No communicated individual asthma action plans

The absence of a specific asthma school plan and lack of clarity about the health role of teachers could delay treatment during asthma attacks. Teachers were unsure of what to do when dealing with children with asthma symptoms, some 'panicked' while several teachers sought help from colleagues considered to have better skills in managing asthma. Both the teachers and the school health team observed that many children with asthma did not bring their reliever inhalers to school. Without a reliever inhaler, teachers tried different strategies when their students had symptoms, for example, relaxation and breathing techniques while waiting for the parents to arrive.

> I was shocked when she was breathless; I am not sure how to help her. I patted her back and massaged her a bit, she was better, but after that, she coughed, and I panicked and quickly called her mother. (D1_FGD 1, teacher)

Participants identified some poor asthma self-management practices on the part of parents. Teachers observed that parents rarely approached the school regarding their child's illness and few had shared their expectations and plans with teachers to support self-management of their child's illness at school (box 1). Even when children had brought an inhaler to school, the teachers had not been asked to supervise its use.

> The child I told you earlier, his parents did not inform us… Only two parents in this school informed us about their child's illness. (HA5_IDI5, teacher)

### Suggested solutions

Participants, particularly those with asthma experience and training, identified a need for a school asthma programme to deliver asthma education and introduce an asthma school plan (box 2). On the other hand, participants without asthma experience or training initially thought a school asthma programme was unnecessary, though some changed their views during the focus group discussions as they became aware of the risk of asthma attacks occurring at school.

> Asthma is not common in school. So, I don't think it (asthma programme) is necessary. But during this discussion, I think teachers should know how to help children with asthma if a child has an attack. It's better. (AN6_FGD 6, a teacher without asthma experience)

To improve the self-management of children with asthma, healthcare professionals and some teachers suggested the school programme could target the children, their parents and teachers as primary school children were not capable of self-managing asthma independently. The children could be taught to improve their self-management skills for example, how to avoid asthma triggers, monitor their symptoms, use a reliever inhaler, and get help if having asthma symptoms/attacks. To support the self-management of the children, their parents and teachers could receive similar asthma education.

> The child themselves need to know, the severity of the symptoms,… the parents should know that the child has asthma… in the school, in case of any emergency, the teachers must know the child's asthma condition and how severe the asthma is (SA4_IDI4, health policymaker)

Participants believed good communication with parents would help school staff to support a child's self-management. Prior agreement on emergency asthma school plans could help boost teachers' confidence in dealing with an attack and protect them from being blamed. A supporting letter from a healthcare professional regarding a child's asthma condition and individual asthma action plan was suggested as another way to provide the information to schools. A few healthcare professionals viewed schools as a platform to bridge communication with parents who did not bring their children for routine reviews.

> We need cooperation from the parent… parents should inform a teacher that 'my child has asthma and need this treatment'. So, the teacher will tell the other teachers and take care of the child… If parents provided a black and white instruction, that is even better. (T6_IDI 6, teacher)

## Implementation challenges

Teachers and members of the school health team were quite negative about school asthma programmes because of the workload (box 2). Teachers were already burdened with academic responsibilities and thought an intensive programme might affect children's learning and general education. The school health team had similar concerns and already felt overwhelmed by their workload, having to provide vaccination, screenings, and general health education in all the schools in the district.

> Cannot, we have a lot of programmes. There is no time to allocate in our schedule. A programme like this, can only be done infrequently. (SA6_FGD6, teacher)

Teachers needed to recognise asthma symptoms/ attacks and provide first aid asthma treatment, but the parents of children with asthma needed to take responsibility for self-management of their children. Parental participation in the programme was thus considered crucial but parents' attendance at school health activities was often very poor either because of work commitments or because they perceived that the health programme was not important for them.

> The lower education group… maybe they are not financially capable, care less about health, maybe work is a priority. So, if we invite parents to come, this group may not have time to attend. (A2_FGD 2, teacher)

The target groups for the school asthma programme have different needs and capabilities. Primary schools covered a wide age range (7–12 years) and the teachers' experience was that lessons needed to be customised to the child's educational level. Some teachers suggested the programme should be delivered in the native language of the children (including Tamil or Chinese as opposed to just the national language) for better understanding.

> HA7: English, Tamil can.
> VA7: But if it is in Tamil, they will understand better, because it is their native language.
> TA7: But some lower and upper primary can understand Malay. (FGD 7, teachers)

## Implementation enablers

Participants suggested simple, engaging, and tailored approaches catering for busy schedules, and different individual needs and capabilities (box 2). The delivery ought to be fun and appealing to children and adults. They suggested quizzes, demonstrations and the use of digital media such as videos to assist sessions' delivery.

> We can use visual aids, we teach first. After that, we ask them to demonstrate. Something like a simulation, if this happened, what will you do? (NO7_IDI 7, teacher)

Healthcare professionals considered good partnerships with schools as an essential enabler to promoting the uptake of school asthma programmes. The school health team had a long-term relationship with the schools and suggested various approaches to building the partnership including involvement of the school administrators at the early stage of the programme development. A good partnership with schools also helped identify and address implementation issues, for example, teachers from Chinese/Tamil schools volunteered to translate the intervention to help children who were not able to understand Malay.

> We have to hold a meeting with the headmaster and the teacher-in-charge of health… once the administrators are interested, they will suggest to you the teacher who will be in charge of the programme… So, you will work with the teacher to develop and run the programme. (RO_FGD 3, school health team)

All participants proposed that healthcare professionals should deliver the intervention in schools, ideally as a multidisciplinary team including pharmacists and paediatricians. To increase parents' participation, teachers suggested wider community involvement such as parent-teacher associations, community organisations and non-governmental organisations to help promote the intervention. Policy-makers highlighted the need for introducing a national asthma school plan to improve asthma care of children during attacks.

> I think this matter will need a clear S.O.P. (standard operating procedure)… Although we have experience with our children, (but) for us to apply the same thing to someone's children, we are worried. (NO8_IDI8, education policymaker)

## DISCUSSION
### Summary of findings

Asthma awareness and experience of managing asthma was limited among the school staff and, with no specific asthma school plan to provide guidance, treatment during asthma symptoms/emergencies could be inappropriate or delayed. School health circular letters were not clear about the teachers' health role in schools. To improve this, participants suggested a school asthma programme that focused on introducing an asthma school plan and improving awareness about asthma among children, their parents and teachers. Communication between healthcare professionals, parents and the school also needed improvement. The participants anticipated heavy workload, poor parental participation and diverse individual capabilities and needs as challenges to implementing school asthma programmes. Tailored approaches, good partnership with schools and multiorganisation involvement were discussed as enablers to develop and deliver the programme. We will use these findings to inform the

development of a school-based asthma programme in Malaysia.

## Study findings in comparison with other studies

Our interviewees suggested developing a school asthma programme involving parents and teachers, that focused on asthma awareness, and an asthma school plan for use in attacks. Similar community-wide programmes have been implemented in other contexts.[31 32] Education for staff and children is a common component,[14 15] and could correct common misperceptions about asthma and asthma medication. Involvement of school health teams could improve the accessibility of healthcare for children with asthma.[19 33–35] School asthma programmes offer the opportunity to focus self-management education on children and involve children's social network to support their self-management skills and practices.[36 37]

The lack of awareness among school staff about which students had asthma, combined with poor understanding of symptoms and first aid treatment leads to hesitancy and delay in taking action risking deterioration or even death.[38 39] Delineating the health roles of teachers in an asthma school plan and providing adequate training for school staff could improve their confidence and prevent treatment delay.[39 40] However, such strategies rely on parents providing asthma reliever medication; without this, school staff will have to call parents or get immediate medical help.[38 41] School asthma guidelines in Australia, Canada, UK and USA encourage inhalers to be labelled and kept with the child or the class teacher in primary schools without a school nurse.[8 42–44] These guidelines also recommend that schools should have an emergency asthma kit to include a reliever inhaler, spacer and agreed information on when and how to use it.[8–10 43] Establishing immediate access to an emergency reliever inhaler in schools required high-level policy changes and detailed guidance for using first aid asthma treatment.[43 45] Therefore, a clear guideline to inform stakeholders for example, policy-makers, school leaders, parents and the community will aid the development and implementation of school asthma programmes.

Highlighting the importance of the programmes to stakeholders particularly policy-makers could ensure successful implementation of school asthma programmes. Policy supporting improvement in school asthma management is a powerful tool in prioritising and providing resources to assist implementation. A good partnership with schools throughout the development of asthma school plans and programmes is crucial to identify challenges and enablers for successful implementation.[46 47] The presence of school plans does not guarantee teachers are equipped with the skills to manage asthma symptoms/attacks.[48] Teachers need to be aware of the school plan, and receive education and training in the use of reliever treatment to equip them to handle asthma events in school.[48 49] Tailoring designs according to the needs and capabilities of individuals and schools could improve the uptake and mitigate the challenges of the

programmes.[19 50] Healthcare providers play a crucial role in the partnership. They need to provide personalised self-management education to a child including assessing if the child can self-carry and self-administer their medication. They can also provide a personalised asthma action plan to be shared with the school.[5 45] Involving the wider community, for example, parent and teacher association, healthcare providers and social services may support delivery and sustainability of school asthma programmes.[19 51 52]

## Study strengths and limitations

This is one of a few studies on school asthma programmes conducted in LMICs. Flexible time, venue and data collection methods facilitated recruitment of a study population with various backgrounds which contributed to the diverse views of participants. Although only three policy-makers were involved in this study, we reached data saturation after the second policy-maker and stopped data collection after the third when no new themes emerged. Participants with less experience on the topic may have contributed less to the focus groups though they were provided with opportunities to provide their views. Nevertheless, the findings may not be wholly applicable in other settings, and an intervention based on our findings will need adaptation for other contexts. Memos, field notes and multidisciplinary team discussions aided analysis and informed interpretation.

We addressed the trustworthiness of our data and analysis. Some loss of meaning may have occurred during the translation of Malay quotes; however, we analysed the data in the source language and checked the semantic and context equivalence for translated quotes. We did not conduct member checks but shared the preliminary analysis with some participants. Thematic analysis may reduce the interpretive details of each participant as the coding process focuses on the content, and less on the processes and phenomena within individuals. However, we used a pragmatic, inductive approach with constant comparisons of data and discussions to inform our understanding of the views expressed. We were aware of the risk of researcher bias in qualitative research, and actively discussed reflexivity in the regular multidisciplinary team discussions which aided analysis and informed interpretation.

## CONCLUSIONS

School asthma programmes could improve the care and support for children with asthma. Aligned with the WHO Global School Health Initiative recommendations,[1] the participants in our study highlighted the importance of tailored approaches, a good partnership with schools and engaging the wider community at an early stage of developing a school asthma programme. There is a need for a clear asthma guideline on school-based programmes to inform policy-makers, school leaders and the community in Malaysia.

**Acknowledgements** We would like to acknowledge all the participants who voluntarily participated in this research and the local authorities for their support during the study. The RESPIRE collaboration comprises the UK Grant holders, Partners and research teams as listed on the RESPIRE website (www.ed.ac.uk/usher/respire), including Nik Sherina Hanafi and Norita Binti Hussein.

**Collaborators** The RESPIRE collaboration comprises the UK Grant holders, Partners and research teams as listed on the RESPIRE website (www.ed.ac.uk/usher/respire), including Nik Sherina Hanafi and Norita Binti Hussein.

**Contributors** All authors (SNR, EMK, SML, SC and HP) contributed to the study conception. SNR performed the data collection, checking and coded the transcripts. All authors (SNR, EMK, SML, SC and HP) contributed to data analysis and interpretation of data. SNR drafted the manuscript and all authors (SNR, EMK, SML, SC and HP) provided critical revisions and editing of the manuscript. SML deceased in December 2021 before submission of the second revision of the manuscript. SNR accepts full responsibility for the work and the conduct of the study, had access to the data, and controlled the decision to publish.

**Funding** This research was commissioned by the UK National Institute for Health Research (NIHR) Global Health Research Unit on Respiratory Health (RESPIRE), using UK Aid from the UK Government.

**Competing interests** None declared.

**Patient consent for publication** Consent obtained directly from patient(s).

**Ethics approval** Ethical approval from the Malaysian National Medical Research and Ethics Committee (NMRR-18-3136-44042 (IIR)).

**Provenance and peer review** Not commissioned; externally peer reviewed.

**Data availability statement** Data are available on reasonable request. Data may be obtained from a third party and are not publicly available. Due to ethical concerns, supporting data cannot be made openly available and are stored under restricted access in the University of Edinburgh DataVault. Further information about the data and conditions for access are available via the University of Edinburgh Research Explorer (https://doi.org/10.7488/191f40bd-ace9-4c8a-a36c-43b796303166). Additional non-disclosive details relating to other aspects of the data are openly available from the University of Edinburgh DataShare repository at (https://doi.org/10.7488/ds/3004)

**ORCID iD**
Siti Nurkamilla Ramdzan http://orcid.org/0000-0002-4427-3778

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
