## [Reviewer comments · BMJ Open]

ARTICLE DETAILS

TITLE (PROVISIONAL)	Stakeholders' views of supporting asthma management in schools with a school-based asthma programme for primary school children: A qualitative study in Malaysia
AUTHORS	Pinnock, Hilary; Ramdzan, Siti Nurkamilla; Khoo, Ee Ming; Liew, Su May; Cunningham, Steven

VERSION 1 – REVIEW

REVIEWER	Isik, Elif Texas Woman's University - Houston Center, College of Nursing
REVIEW RETURNED	23-May-2021

GENERAL COMMENTS	Thank you for this research study. Introduction section: You must add more information related to asthma morbidity and mortality rates in Malasia for school age children. After that, you should inform the readers for the 7-12 years old children asthma morbidity and mortality rate. Explain the school health team-this team includes (who)? Are they teachers, secretary, or? Explain the focus group. Each focus group involved who? (For example, one focus group involved 3 teachers, 1 healthcare professional and 2 policy makers). You need to inform the readers who were in each focus group. Discussion section: You should discuss about self-carry quick relief asthma medication and also you need to discuss about administering asthma medication. For that reason, you need to discuss generic and individual asthma action plan and physican and parent consent form for self-administering or adminisitering asthma medication at schools. Line 197-198: You wrote "children with asthma must limit their physical activity', and 'asthma attacks can be anticipated' as misconceptions, but they are not totally misconception. If there is suggested no physical activity-this is a real misconception. Also, anything can trigger asthma attacks, so the other one also is not misconception. Your quote did not support these misconceptions. Line 323: You are talking about death because of delayed action- do you have any data to support this notion? Did death or ER happened in last 10- years because of asthma while students at school? if yes, please support it with data. In discussion and conclusion section, you need to suggest clear asthma guidelines to inform policy members, school leaders, and the community. Also, You need to mention educating the children for asthma symptoms, triggers, and manage their condition at doctor visits if students can self-carry and self-administer their medicine. Generic plan must be known by school health team and Iso teachers for asthma. However, students with asthma should have
--

	a written asthma action plan signed by physician and parent. This action plan must be provided by parent for the teachers and school health team. This action plan must be reviewed at school with parents, school health team, and teachers to prevent negative outcomes. Instead using of reliever medication, I suggest using quick-relief asthma medications or quick-relief asthma inhalers. Since, relievers can be long-term or short-term (quick-relief). I recommend using an editor for proofread for this manuscript, since a few sentences can be written in a better way.
--	--

REVIEWER	Zairina, Elida Universitas Airlangga, Pharmacy Practice
REVIEW RETURNED	01-Jul-2021

GENERAL COMMENTS	The paper describes schools staff, healthcare professionals, and policymakers about asthma and school-based asthma management regarding developing a primary school asthma programme in Malaysia. Overall, this is a good paper. Although the qualitative study designed was not aimed to be generalized to the whole population, it would be better if the authors can describe the next plan based on the study's result is? Any intervention studies needed? And is there any particular reason why the parents were not involved in the interview? What is the prevalence of children with asthma in Malaysia? Kindly add it into the introduction section, so the readers will have clear pictures of why a primary school asthma program in Malaysia is essential. Because the study was designed qualitatively, how will the findings be applied in other areas of Malaysia? There was no mention of where or when the participants were recruited.
--

REVIEWER	Sonney, Jennifer University of Washington, Child, Family, and Population Health Nursing
REVIEW RETURNED	09-Jul-2021

GENERAL COMMENTS	Consider clarifying the objective section of the abstract - I would recommend including that there is no national asthma program in Malaysia, which provides a rationale for the aim of the study. - Methods- It would be useful to further describe the ways in which the study team established trustworthiness. I saw mention of constant comparison, but would appreciate it if this section were further developed. - Limitations - should potential researcher bias be included in limitations? - Overall - there are numerous grammatical errors, the paper would benefit from an editorial review.
--

VERSION 1 – AUTHOR RESPONSE

Comments from reviewer 1

3) Introduction section: You must add more information related to asthma morbidity and mortality rates in Malaysia for school age children. After that, you should inform the readers for the 7-12 years old children asthma morbidity and mortality rate.

There is lack of detailed data on asthma morbidity and mortality rate in Malaysian children by different age groups. We added the more general information on the prevalence and asthma control among children in Malaysia on page 5, line 97-101:

“In Malaysia, an estimated 9% of children have asthma^{19 20}, at least half of whom are poorly controlled.^{19 21} Despite this morbidity, the national school health service guideline does not include an asthma programme²², and a qualitative study conducted among primary school children with asthma and their parents revealed a perception of poor support for asthma care in schools.^{23 24}”

4) Explain the school health team-this team includes (who)? Are they teachers, secretary, or?

We have clarified and added details of school health team on page 6, 118-121:

“In these schools, there are no school nurses and the healthcare of school children is provided by the Ministry of Health in the form of a school health team comprised of family medicine specialists, medical officers and nurses, who provide general health screening and education in each district.²²”

5) Explain the focus group. Each focus group involved who? (For example, one focus group involved 3 teachers, 1 healthcare professional and 2 policy makers). You need to inform the readers who were in each focus group.

We have added a table (table 2) with details of the participants in each focus group.

6) Discussion section: You should discuss about self-carry quick relief asthma medication and also you need to discuss about administering asthma medication. For that reason, you need to discuss generic and individual asthma action plan and physician and parent consent form for self-administering or administering asthma medication at schools.

Thank you for your suggestions. We had discussed generic action plans on page 19, line 364-368, but have now highlighted that need for an emergency inhaler in schools.

“School asthma guidelines in Australia, Canada, United Kingdom and United States encourage inhalers to be labelled and kept with the child or the class teacher in primary schools without a school nurse.^{7 42-44} These guidelines also recommend that schools should have an emergency asthma kit to include a reliever inhaler, spacer and agreed information on when and how to use it.^{7-9 43}”

We have also added the following sentence (on page 20, line 385-388) to highlight the role of the child’s healthcare advisor and the need to specify whether a child is able to self-medicate.

“Healthcare providers play a crucial role in the partnership. They need to provide personalised self-management education to a child including assessing if the child can self-carry and self-administer their medication. They can also provide a personalised asthma action plan to be shared with the school.^{5 45}”

7) Line 197-198: You wrote 'children with asthma must limit their physical activity', and 'asthma attacks can be anticipated' as misconceptions, but they are not totally misconception. If there is suggested no physical activity-this is a real misconception. Also, anything can trigger asthma attacks, so the other one also is not misconception. Your quote did not support these misconceptions.

Thank you for highlighting the mismatch between the text and the supporting quotes. We have now added a quote and rephrased the sentence in the text to illustrate the quotes better (on page 11, line 219-220):

- We have added a quote to illustrate the misconception that children with asthma should not do any exercise. We have changed the expression ‘limiting physical activity’ in the text to ‘cannot do physical activities’ to reflect the quote better.

- The other quote illustrates that the staff member was surprised to learn that an attack could start suddenly (perhaps as a result of meeting a trigger) in a child who was well. We have changed the text to clarify the misconception that ‘asthma attacks do not occur suddenly in a well child’.

8) Line 323: You are talking about death because of delayed action-do you have any data to support this notion? Did death or ER happened in last 10- years because of asthma while students at school? if yes, please support it with data.

Although, there are no data from Malaysia on asthma deaths in schools, highly publicised asthma deaths in UK schools specifically criticised the delay in seeking medical attention [1], and resulted in initiatives to reduce delays in managing asthma in schools including changes in the law to allow teachers to administer reliever treatment whilst seeking medical help. This echoes the findings of previous studies implicating delay as a factor in asthma deaths in schools (ref 39 and ref 40). Both these references are now cited in the text on page 19, line 359.

9) In discussion and conclusion section, you need to suggest clear asthma guidelines to inform policy members, school-leaders, and the community.

We have added this suggestion in the discussion, page 19, line 371-373-.

“Therefore, a clear guideline to inform stakeholders e.g. policymakers, school leaders, parents and the community will aid the development and implementation of school asthma programmes.”

This was also added in conclusions, page 21 line 419-421.

“There is a need for a clear asthma guideline on school-based programmes to inform policymakers, school leaders, and the community in Malaysia.”

10) Also, You need to mention educating the children for asthma symptoms, triggers, and manage their condition at doctor visits if students can self-carry and self-administer their medicine. Generic plan must be known by school health team and also teachers for asthma. However, students with asthma should have a written asthma action plan signed by physician and parent. This action plan must be provided by parent for the teachers and school health team.

This action plan must be reviewed at school with parents, school health team, and teachers to prevent negative outcomes.

We completely agree about the role of the child’s own healthcare advisor and have added a statement about individual action plans and the ability of the child to self-medicate in response to the comment above. This now reads (Page 20, Line 385-388):

“Healthcare providers play a crucial role in the partnership. They need to provide personalised self-management education to a child including assessing if the child can self-carry and self-administer their medication. They can also provide a personalised asthma action plan to be shared with the school.5 45”

We also agree that school staff need to be aware of the existence of a generic school plan and have now added ‘awareness’ to our statement about the need to provide education and training for teachers. The sentence on (page 20, line 379-383) now reads:

“The presence of school plans does not guarantee teachers are equipped with the skills to manage asthma symptoms/attacks.48 Teachers need to be aware of the school plan, and receive education and training in reliever treatment to equip them to handle asthma events in school.48 49”

11) Instead using of reliever medication, I suggest using quick-relief asthma medications or quick-relief asthma inhalers. Since, relievers can be long-term or short-term (quick-relief).

We have considered this suggestion but would prefer to use the term ‘reliever’ as this is in line with global guidelines (GINA), Malaysian guidelines, and both BTS/SIGN and NICE in the UK. 12) I recommend using an editor for proofread for this manuscript, since a few sentences can be written in a better way.

Authors who are native English speakers have proofread the manuscript and made some changes in a few sentences to make it clearer.

Comments from reviewer 2

13) Overall, this is a good paper. Although the qualitative study designed was not aimed to be generalized to the whole population, it would be better if the authors can describe the next plan based on the study's result is? Any intervention studies needed?

Thank you.

We have added a short statement about our future plans based on the study findings in the discussion section on page 18 line 344-345:

"We will use the findings to inform the development of a school-based asthma programme in Malaysia."

14) And is there any particular reason why the parents were not involved in the interview?

We had conducted an earlier qualitative study on primary school children with asthma and their parents. This is described on page 5, line 98-105.

"Despite this morbidity, the national school health service guideline does not include an asthma programme 23, and a qualitative study conducted among primary school children with asthma and their parents revealed a perception of poor support for asthma care in schools.^{24 25} Building on the findings of the qualitative study^{24 25}, we aimed to explore the views of school staff, healthcare professionals and policymakers about asthma and school-based asthma management to inform the development of an intervention on primary school asthma programme in Malaysia."

15) What is the prevalence of children with asthma in Malaysia? Kindly add it into the introduction section, so the readers will have clear pictures of why a primary school asthma program in Malaysia is essential.

We have now added the information on page 5, line 97-101.

"In Malaysia, an estimated 9% of children have asthma^{20 21}, at least half of whom are poorly controlled.^{20 22} Despite this morbidity, the national school health service guideline does not include an asthma programme 23, and a qualitative study conducted among primary school children with asthma and their parents revealed a perception of poor support for asthma care in schools.^{24 25}"

16) Because the study was designed qualitatively, how will the findings be applied in other areas of Malaysia?

In order to help readers judge the applicability of our findings to other settings, we have described the context of our study on page 6. Building on global literature [2], the findings of this qualitative study will inform the development of the core components of a primary school asthma programme in Malaysia, whilst allowing for adaptation and tailoring for implementation in other settings. We have added a sentence to the limitations (page 20, line 397-399)

"Nevertheless, the findings may not be wholly applicable in other settings, and an intervention based on our findings will need adaptation for other contexts"

17) There was no mention of where or when the participants were recruited.

The date when the study was conducted is provided at the beginning of the methods (between May and December 2019). We have now added information about where we recruited participants on page 6, line 121-125:

"We conducted the study in the central region of Peninsular Malaysia and purposively sampled school staff (teachers and administrative staff) from five government schools (two Malay, two Chinese and one Indian), to represent the three main ethnic groups and diverse cultures in Malaysia."

Comments from reviewer 3

18) Consider clarifying the objective section of the abstract - I would recommend including that there is no national asthma program in Malaysia, which provides a rationale for the aim of the study.

We added information as suggested which reads,

“Despite WHO recommendations, no asthma programme is included in the Malaysian national school health service guideline.”

19) Methods- It would be useful to further describe the ways in which the study team established trustworthiness. I saw mention of constant comparison, but would appreciate it if this section were further developed.

We added further descriptions regarding rigour of the methods page 7-8, line 156-160.

“SNR read the transcripts multiple times for data familiarisation and developed initial codes according to the research question. Constant comparison and merging of initial codes allowed the development of sub-codes, sub-themes and themes, and we then constructed a schema to describe the relationship between themes.”

20) -Limitations - should potential researcher bias be included in limitations?

We now include a statement which reads,

“We were aware of the risk of researcher bias in qualitative research, and actively discussed reflexivity, in the regular multidisciplinary team discussions which aided analysis and informed interpretation.” (page 20, line 408-410)

21) Overall - there are numerous grammatical errors, the paper would benefit from an editorial review. Authors who are native English speakers have proofread the manuscript and made some grammatical changes.

VERSION 2 – REVIEW

REVIEWER	Sonney, Jennifer University of Washington, Child, Family, and Population Health Nursing
REVIEW RETURNED	29-Oct-2021

GENERAL COMMENTS	Thank you for the opportunity to review "Stakeholders' views of asthma and school-based asthma programme for primary school children: A qualitative study in Malaysia". This study used a qualitative design, specifically semi-structured interviews, to explore the asthma knowledge of school staff, healthcare professionals and policymakers engaged in programs for school-age children. This is an important area of study and the authors make a strong argument for the need for a school-based asthma program. My comments aim to strengthen/clarify a few areas. My primary suggestion is to consider clarifying the study aims. "views of school staff, healthcare professionals and policymakers" is nonspecific and it seems as though the study really was a needs assessment that explored barriers to asthma management in the schools, no? Abstract: * While the diverse participant pool is interesting, it is difficult for me as a reader to clearly understand the overall study aim. * Specific study design (qualitative descriptive, for example, is not included). * Consider clarifying what "first-aid asthma treatments" means or rephrasing. INTRODUCTION
---

I recommend that you consider resequencing some of your introduction for improved flow for your reader. You open with the WHO global initiative, which is fine. Paragraph 2 opens with asthma being common, but ends with WHO school-based asthma program recommendations. Paragraph 3 again touches on individual asthma symptoms again, but then goes back to WHO recommendations. I think the flow could be better if each paragraph had a specific topic sentence/focus, such as:

- 1 - WHO initiative
- 2 - significance - asthma prevalence, symptoms, consequences of poor asthma management. Statistics would be helpful, but a focused discussion on why asthma must not be ignored would be helpful.
- 3 - School-based asthma programs - WHO call for such programs, discussion of those that are successful, then the gap in Malaysia. Are there key components of a school-based program? Communication with parents, asthma action plan, medications at school, etc? This information would provide context for the readers as they review your results, which highlight the need for some of the items listed.
- 4- Purpose paragraph with study aims.

Additional comments on introduction:

* Consider clarifying page 6 line 84 "A school asthma programme offers an exemplar for school health NCD programmes.." - I am not certain what this means.

* You go on to state that asthma programs are specifically recommended by the WHO - it would seem you could provide a rationale as to why (worsened asthma outcomes/other consequences)

METHODS

* I am accustomed to more detail on study design (not just qualitative).

* How were prospective participants approached/recruited? Specifically, how were they invited? How were they consented?

* Consider addressing trustworthiness

* Consider describing the demographic form (# items, variables collected) and semi-structured interview guides (#items, focus of questions). Were the interview guides the same regardless of participant perspective/profession? The reader needs to know this, not just look for them in supplementary materials.

RESULTS

*Page 13 line 35 - what are "circulars"?

* I wonder if the use of "self-management" is appropriate, such as in page 15 line 70, given that the authors go on to state that "primary school children were not capable of self-managing asthma independently". Would "asthma management practices" or some variation be more appropriate? This pertains to the entire manuscript.

* I find some of the results relate well to what might be included in a school-based asthma program, but it would be helpful for the readers if you outlined what is typically included. For example, I did not expect page 17 line 04+, which appeared to call for educational interventions in school. I think this is a great idea, but is this typically part of a school-based asthma program? I made the assumption that such programs focused on paperwork and coordination between parent and school (asthma action plan,

	medications at school, notifications etc). Are the authors suggesting that all students with asthma participate in such an intervention? DISCUSSION Again, early framework outlining what constitutes a school-based asthma program is necessary to understand and follow the discussion. I question whether another limitation is the minimal involvement of "policymakers".
--	---

VERSION 2 – AUTHOR RESPONSE

Comments from reviewer 3

1) Thank you for the opportunity to review "Stakeholders' views of asthma and school-based asthma programme for primary school children: A qualitative study in Malaysia". This study used a qualitative design, specifically semi-structured interviews, to explore the asthma knowledge of school staff, healthcare professionals and policymakers engaged in programs for school-age children. This is an important area of study and the authors make a strong argument for the need for a school-based asthma program. My comments aim to strengthen/clarify a few areas.

Thank you for your feedback which improved our manuscript.

2) My primary suggestion is to consider clarifying the study aims. "views of school staff, healthcare professionals and policymakers" is nonspecific and it seems as though the study really was a needs assessment that explored barriers to asthma management in the schools, no?

We revised our description of the study aims in the abstract; page 2, line 31-33, which now reads: "Therefore, we aimed to explore the views of school staff, healthcare professionals and policymakers about the challenges of managing asthma in schools and the potential of a school asthma programme for primary school children."

The aim in the text (end of introduction; page 6) is stated as "We therefore aimed to explore the views of school staff, healthcare professionals and policymakers about the challenges of managing asthma in school, and the potential of a school asthma programme for primary school children to improve care."

We also rephrased the title of the manuscript to, "Stakeholders' views of supporting asthma management in schools with a school-based asthma programme for primary school children: A qualitative study in Malaysia"

Abstract:

3) * While the diverse participant pool is interesting, it is difficult for me as a reader to clearly understand the overall study aim.

We have now clarified our aim in the response to the previous comment in the title (page 1), abstract (page 2) and main text page 5).

4) * Specific study design (qualitative descriptive, for example, is not included).

We included specific study design on the abstract, page 2, line 34-35, "A focus group and individual interview qualitative study using purposive sampling of participants to obtain diverse views"

5)* Consider clarifying what "first-aid asthma treatments" means or rephrasing.

We rephrased the sentence on page 2, line 42 to, "School staff had limited awareness of asthma and what to do in emergencies."

INTRODUCTION

6) I recommend that you consider resequencing some of your introduction for improved flow for your reader. You open with the WHO global initiative, which is fine. Paragraph 2 opens with asthma being common, but ends with WHO school-based asthma program recommendations. Paragraph 3 again touches on individual asthma symptoms again, but then goes back to WHO recommendations. I think the flow could be better if each paragraph had a specific topic sentence/focus, such as:

1 - WHO initiative

2 - significance - asthma prevalence, symptoms, consequences of poor asthma management. Statistics would be helpful, but a focused discussion on why asthma must not be ignored would be helpful.

3 - School-based asthma programs - WHO call for such programs, discussion of those that are successful, then the gap in Malaysia. Are there key components of a school-based program? Communication with parents, asthma action plan, medications at school, etc? This information would provide context for the readers as they review your results, which highlight the need for some of the items listed.

4- Purpose paragraph with study aims.

We restructured paragraphs 2-4 of the introduction as suggested (page 4-6) line 81-110.

Asthma is the commonest NCD in children with a global prevalence of 11.6%. Children with poor asthma control have symptoms of wheeze, cough and difficulty breathing that affect their daily activities. Good self-management of asthma improves symptoms and reduces the risk of attacks, but for primary school children, self-management of asthma requires support from their social network including their parents, school and teachers. Young children can spend up to half their waking hours in schools, and school staff could help students develop asthma self-management skills. Furthermore, attacks can occur unexpectedly, and schools need plans for responding appropriately and promptly to asthma symptoms.

The WHO endorses the role of school in supporting children with asthma and recommends that asthma should be an essential school health service in all countries. School-based asthma self-management educational interventions that were theory-driven, involved parent, included children satisfaction and done outside children's free time have been shown to improve asthma control, reduce school absenteeism and asthma attacks, though few were conducted in LMICs. These

programmes are complex interventions that can be successful if tailored to the target population, and adapted to the context. The WHO has also highlighted the importance of policy and organisation-level collaboration and engagement with parents, students and teachers.

In Malaysia, an estimated 9% of children have asthma, at least half of whom are poorly controlled. Despite this morbidity, the national school health service guideline does not include an asthma programme, and prior qualitative study conducted among primary school children with asthma and their parents revealed a perception of poor support for asthma care in schools.

We therefore aimed to explore the views of school staff, healthcare professionals and policymakers about the challenges of managing asthma in schools and the potential of a school asthma programme for primary school children to improve care.

Additional comments on introduction:

7) * Consider clarifying page 6 line 84 "A school asthma programme offers an exemplar for school health NCD programmes.." - I am not certain what this means.

We have deleted this sentence in our revised introduction (see previous question).

8) * You go on to state that asthma programs are specifically recommended by the WHO - it would seem you could provide a rationale as to why (worsened asthma outcomes/other consequences)

In our revised introduction, this sentence now follows a description of the potential roles of school (in helping students develop asthma self-management skills and also being able to respond appropriately and promptly to asthma emergencies). We then state that the WHO endorses these roles as a rationale for asthma being an essential school health service in all countries.

The text (page 5, line 86-93) now reads:

"Young children can spend up to half their waking hours in schools, and school staff could help students develop asthma self-management skills. Furthermore, attacks can occur unexpectedly, and schools need plans for responding appropriately and promptly to asthma symptoms. The WHO endorses the role of school in supporting children with asthma and recommends that asthma should be an essential school health service in all countries."

METHODS

9)* I am accustomed to more detail on study design (not just qualitative).

We have included additional information to describe the study design on page 6, line 113-117, "This mixed focus group and individual interview qualitative study was conducted between May and December 2019 with ethical approval from the Malaysian National Medical Research and Ethics Committee (NMRR-18-3136-44042 (IIR)) and sponsorship from the Academic and Clinical Centre for Research and Development, University of Edinburgh (AC18113)."

10) * How were prospective participants approached/recruited? Specifically, how were they invited? How were they consented?

We have added a statement to explain how the participants were approached on page 7, line 133-134, "Potential participants were invited individually or via their organisation by providing them with a participant information sheet."

Information on consent is provided on page 6, line 118-119 in the context of describing ethical considerations, "All participants provided their written informed consent."

11)* Consider addressing trustworthiness

Although we did not use this terminology, we had addressed trustworthiness in a number of ways which we now discuss under strengths and weakness. The paragraph (page 22, line 403-413) now reads:

"We addressed the trustworthiness of our data and analysis. Some loss of meaning may have occurred during the translation of Malay quotes; however, we analysed the data in the source language and checked the semantic and context equivalence for translated quotes. We did not conduct member checks but shared the preliminary analysis with some participants. Thematic analysis may reduce the interpretive details of each participant as the coding process focuses on the content, and less on the processes and phenomena within individuals. However, we used a pragmatic, inductive approach with constant comparisons of data and discussions to inform our understanding of the views expressed. We were aware of the risk of researcher bias in qualitative research, and actively discussed reflexivity in the regular multidisciplinary team discussions which aided analysis and informed interpretation."

12)* Consider describing the demographic form (# items, variables collected) and semi-structured interview guides (#items, focus of questions). Were the interview guides the same regardless of participant perspective/profession? The reader needs to know this, not just look for them in supplementary materials.

We used supplementary files to avoid exceeding the word count and number of tables/figures allowed for the manuscript. We have now explained that the variables collected are reported in Table 1 and we have added examples of the variables in the methods section. We have also added the information that different topic guides that were used for different professions. The text (on page 7, line 138-144) now reads:

"We collected basic socio-demographic data, e.g. age, gender, profession and asthma experience (Supplementary 1, and with variables reported in table 1), and developed piloted semi-structured topic guides (Supplementary 2) based on the biopsychosocial model and previous literature to stimulate discussion. Different topic guides containing specific questions relevant to different roles were used for different professions."

RESULTS

13) *Page 13 line 35 - what are "circulars"?

We rephrased 'circulars' to 'general advice (e.g. in circular letters from authorities) to describe documents used in the schools to guide school staff on their standard operating procedure or practices.

14)* I wonder if the use of "self-management" is appropriate, such as in page 15 line 70, given that the authors go on to state that "primary school children were not capable of self-managing asthma independently". Would "asthma management practices" or some variation be more appropriate? This pertains to the entire manuscript.

We use the term in accordance with global and local asthma management guidelines¹⁻³ to describe the decisions and actions people (in this context children and their parents) take to cope with their condition. Although clearly the degree of autonomy evolves throughout childhood and adolescence, our previous work highlights how even primary school children begin to experiment with self-management albeit strongly influenced by parents.⁴ Our revised introduction clarifies that self-management in children 'requires support from their social network including their parents, school and teachers'.

15)* I find some of the results relate well to what might be included in a school-based asthma program, but it would be helpful for the readers if you outlined what is typically included. For example, I did not expect page 17 line 04+, which appeared to call for educational interventions in school. I think this is a great idea, but is this typically part of a school-based asthma program? I made the assumption that such programs focused on paperwork and coordination between parent and school (asthma action plan, medications at school, notifications etc). Are the authors suggesting that all students with asthma participate in such an intervention?

The aim of the study was to explore different views and ideas about the content of school-based asthma programmes to inform the development of an intervention. The importance of including an educational intervention to all students with asthma emerged from the findings. In fact, this aligns with all school-based interventions described in the Cochrane review and our systematic review which include education.^{5,6} Therefore, we will include the findings to developing a school-based programme together with findings from a systematic review and a previous qualitative study. We now emphasise (page 19, line 347-348) that education is a common component of school-based programmes. The text reads: "Education for staff and children is a common component, and could correct common misperceptions about asthma and asthma medication."

DISCUSSION

16) Again, early framework outlining what constitutes a school-based asthma program is necessary to understand and follow the discussion.

We have now listed the common components of a school-based asthma programme in the introduction. The text (page 5, line 93-96) now reads:

School-based asthma self-management educational interventions that were theory-driven, involved parent, included children's satisfaction and done outside children's free time have been shown to improve asthma control, reduce school absenteeism and asthma attacks.

17) I question whether another limitation is the minimal involvement of "policymakers".

We added this in limitations on page 21, line 394-396, "Although only three policymakers were involved in this study, we reached data saturation after the second policymaker and stopped data collection after the third when no new themes emerged."

We would like to thank the reviewers for their comments in improving our paper. After adding the reviewer's comments, our manuscript slightly exceeded the maximum word count (word count: 4023). The manuscript has been seen and approved by all authors.

We hope that our revised manuscript meets your approval.